# Tissue-specific synergistic bio-priming of pepper by two *Streptomyces* species against *Phytophthora capsici*

**Sakineh Abbasi[1], Naser Safaie[1]*, Akram Sadeghi[2]*, Masoud Shamsbakhsh[1]**

**1** Department of Plant Pathology, Faculty of Agriculture, Tarbiat Modares University, Tehran, Iran,
**2** Department of Microbial Biotechnology, Agricultural Biotechnology Research Institute of Iran (ABRII), Agricultural Research, Education and Extension Organization (AREEO), Karaj, Iran

* nsafaie@modares.ac.ir (NS); aksadeghi@abrii.ac.ir (AS)

**Data Availability Statement:** All relevant data are within the manuscript and its Supporting Information files.

## Abstract

Among several studied strains, *Streptomyces rochei* IT20 and *S. vinaceusdrappus* SS14 showed a high level of inhibitory effect against *Phytophthora capsici*, the causal agent of pepper blight. The effect of two mentioned superior antagonists, as single or combination treatments, on suppression of stem and fruit blight diseases and reproductive growth promotion was investigated in pepper. To explore the induced plant defense reactions, ROS generation and transcriptional changes of selected genes in leaf and fruit tissues of the plant were evaluated. The plants exposed to the combination of two species responded differently in terms of $H_2O_2$ accumulation and expression ratio of *GST* gene compared to single treatments upon pathogen inoculation. Besides, the increment of shoot length, flowering, and fruit weight were observed in healthy plants compared to control. Likely, these changes depended on the coordinated relationships between *PR1*, *ACCO* genes and transcription factors *WRKY40* enhanced after pathogen challenge. Our findings indicate that appropriate tissue of the host plant is required for inducing *Streptomyces*-based priming and relied on the up-regulation of *SUS* and differential regulation of ethylene-dependent genes.

## Introduction

*Phytophthora* blight caused by *Phytophthora capsici* Leonian is one of the devastating soil-borne diseases of pepper (*Capsicum annuum* L.) and several important crops throughout the world [1, 2]. This fungus-like pathogen can display numerous symptoms depending on a host, including foliar blighting, damping-off, wilting, stem, and fruit blight [3]. Using chemical pesticides has developed concerns over environmental health. Biological control programs have become an eco-friendly manner to manage soil-borne diseases and reduce chemical treatments [4, 5]. Some species of *Streptomyces*, gram-positive filamentous bacteria, are considered as plant growth-promoting rhizobacteria (PGPR) colonizing rhizosphere/plant root, and improving plant health and growth. The single application of *Streptomyces* sp. as a biocontrol agent was investigated against some soil-borne fungal pathogens such as *Rhizoctonia solani* [6] and *Fusarium oxysporum* f. sp. *lycopersici* [7]. There is a growing interest in using PGPRs

**Funding:** AS received fund. The grant numbers: 12-05-05-034-09454-950941 Agricultural Biotechnology Research Institute of Iran (ABRII) http://www.abrii.ac.ir/en/ The funder played NO role in study design, data collection and analysis decision to publish or preparation of the manuscript.

**Competing interests:** The authors have declared that no competing interests exist.

capable of solubilizing inorganic phosphorus (P) [8]. Co-inoculation P-solubilizing rhizobacteria and *Rhizobium* led to higher plant growth than their single inoculation [9, 10]. Co-inoculation of biocontrol agents displayed a more pronounced impact on the microbial structure of rhizosphere than a single application [11]. Some studies have evaluated the effects of *Actinomycetes* consortium from either the same or different taxonomic groups [12, 13, 14, 15]. The research programs have mostly focused on biocontrol of damping-off caused by *Phytophthora* species in the seedling stage (root and crown rot), with minimal consideration about fruit blight disease on pepper caused by *P. capsici* [16, 17]. It has been proven that biocontrol agents directly prevent pathogen progress using cell wall degrading enzymes or stimulation of host plant immune defense [18]. Chemical elicitor β-aminobutyric acid (BABA) induces plant systemic defense response against later infection, symbolizing a usable option for *Phytophthora* diseases management [19, 20]. Many plant defense strategies including the production of reactive oxygen species (ROS), mainly $^1O_2$, $H_2O_2$, $O^{\bullet-}2$, and $OH^\bullet$, are inducible and launched in response to stress factors such as pathogen and/or beneficial microbes [21, 22]. ROS generation may lead to different transcriptional responses in salicylic acid (SA) and/ or ethylene (ET) signaling pathways in plants treated with a combination of *Streptomyces* species. Besides, the patterns of pathogenesis-related proteins (PR) and induced systemic resistance (ISR) could be distinct in responsive tissues. Hence, it is important to characterize the responses of the host co-inoculated against *P. capsici* in two plant tissues through gene expression profiles which will offer evidence to the physiological and transcriptional changes during interactions. The aims of this study were to 1) screen of P-solubilizing and antagonist strains of *Streptomyces* against *P. capsici* from microbial culture collection isolated from rhizosphere soils of vegetable greenhouses 2) evaluate the potential of two superior species in improving the vegetative and reproductive growth, and bio-suppression against *Phytophthora* blight disease of pepper by single or combination treatments in greenhouse condition 3) display the changes in host plant in terms of $H_2O_2$ accumulation and gene expression patterns in leaves and fruits against *P. capsici* 4) measure disease index on fruits pre-treated with *P. capsici* and/or with biocontrol *Streptomyces* in single or mixed application after zoospore inoculation.

## Material and method

### Microorganisms

In this research, 106 *Streptomyces* strains were selected from microbial culture collection of ABRII (ABRIICC). It is noteworthy that these bacterial strains were screened based on hydrolytic enzyme activities and antifungal ability against some soil-borne fungal pathogens. According to our previous investigations, these strains were able to grow on nitrogen-free medium and produce siderophore [7]. Eight of these strains produced cellulase (the greatest halo zone/colony diameter ratio for cellulase activity). Nearly complete 16S rRNA gene sequences (1400 nt) obtained by DNA amplification of two selected strains *S. rochei* IT20 and *S. vinaceusdrappus* SS14 were deposited in NCBI GenBank under accession numbers MK858186 and MH041316, respectively. The *Oomycete* pathogen was isolated from pepper plants displaying disease symptoms (Alborz, Iran) and the pathogenicity test was conducted using plug inoculation on pepper seedlings. The pathogen was re-isolated from the necrotic tissue margins of a symptomatic plant. DNA extraction was done by the method previously described [23–25] and polymerase chain reaction (PCR) conducted with universal pair primers (ITS4-ITS5). ITS4-ITS5 nucleotide sequence obtained by DNA amplification of pathogen was deposited in NCBI GenBank under accession number MG670447. Blast analysis showed that this isolate had 99% similarity to *P.capsici*.

## Screen of strains

The spore suspension of *Streptomyces* strains cultured in International Streptomyces Project (ISP2) broth supplemented with 10 mM NaCl (20 μL of the $10^8$ cfu/mL sterile saline solution) were inoculated linearly at the two contrary sides (1 cm from the plate edge) of potato dextrose agar (PDA) plates and incubated for 48 h at 29˚C. A plug (0.5 cm diameter) of *P.capsici* was inoculated at the center of PDA plates [26]. After four days of incubation, the percent inhibition of mycelial growth and germination were calculated using the formula $[(c − t)/c] \times 100$, where 'c' is the pathogen growth radius of a control (in cm) and 't' is the distance of the pathogen growth in the direction of bacteria (in cm).

## Plant growth condition and bacterial treatments

Sterilized seeds of bell pepper (*C. annuum* L. cv Elmas) were placed into an 84-cell plug tray ($50 \times 30 \times 5$ cm deep) filled with peat moss, with one seedling per cell. The seedlings were watered every day with tap water and kept in a greenhouse at 27±2˚C and 16 h brightness/8 h darkness. After 36 days, the seedlings were transferred to pots ($20 \times 35$ cm$^2$) filled with a sterile mixture of field soil and peat moss (2:1). Maximum air temperature on the day of inoculation was 25±3˚C during the trial. For bacterial treatments, *Streptomyces* spores cultured in ISP2 medium were pelleted (1 g fresh weight) by centrifugation (8000 rpm for 10 min) and re-suspended in 10 mL sterile saline solution. The bacterial suspension was added to autoclaved sand (100 g). Five gram of sand containing bacteria was added to the hole of each cultivated plant. Sterilized sand was used as a control. The plants were irrigated until soil saturation and subsequently irrigated throughout the experiment, usually twice a day (80 ml).

## Greenhouse trials

After seven days of bacterial treatment, plants were inoculated with $2 \times 2$ cm$^2$ of five-day-old fungus-like plugs on PDA and applied at 1 cm of the crown of each plant [17]. The treatments included control (mock inoculation), positive control (inoculated with *P.capsici*), soil drenched with BABA as ISR inducer (Sigma-Aldrich), two strains of *Streptomyces* species SS14, IT20, and IT20+ SS14 (combination in amounts of 0.5+0.5) inoculated or non-inoculated with *P.capsici*. The plant growth parameters including shoot length, flowering day, number of flowers and fruits, fruit length, plant and fruit weight were measured and recorded from 30 to 80 days after treatment. Also, disease severity (DS) and disease progress were assessed on a scale from 0 to 5: 0 = no symptoms = 0%, 1 = leaf yellowing = 25%, 2 = minor stem necrosis = 50%, 3 = moderate stem necrosis and some leaf wilt = 75%, 4 = severe stem necrosis and severe wilt, 5 = plant death = 100% [27]. Soil drench with chemical inducer BABA (10mM) was performed 48 h before *P.capsici* (*PC*) inoculation. After seven days of *PC* /mock inoculation leaves and fruits (by length 1 cm) were harvested, frozen in liquid nitrogen and kept at -70˚C for $H_2O_2$ production and gene expression analysis.

## Hydrogen peroxide quantification

The concentration of $H_2O_2$ was measured on lyophilized tissue crushed in 0.1% cold trichloroacetic acid (TCA) and then centrifuged at 12000 rpm for 15 min in a refrigerated centrifuge. The supernatant (0.5 mL) was added to 10 mM phosphate buffer (pH 7.0) and 1 M of iodate potassium (KI) solution and the absorbance was measured at 390 nm [28, 29]. The amount of hydrogen peroxide was calculated using a standard curve.

## RNA extraction and quantitative real-time PCR analysis

Total RNA was extracted using TRIzol reagent (Invitrogen, USA) from leaves and fruits of four biological replicates of each treatment. cDNA was synthesized using 1 μg of each RNA sample after treating with RNase-free DNase I (Invitrogen) using iScript cDNA synthesis kit (BioRad) according to the manual description. Quantitative PCR was performed in a 25 μL reaction containing 1 μL of template cDNA, 0.5 μL of 10 pM of each forward and reverse specific primers designed in this study (Table 1) and iQ SYBR Green Supermix kit (BioRad) on Roche Light Cycler® 96 System real-time PCR. The PCR profile included an initial denaturation step at 95˚C for 3 min, followed by 40 cycles of denaturation (95˚C/ 30 sec), primer annealing (55˚C/45 sec) and primer elongation (72˚C/30 sec), by a final elongation step (72˚C/ 3 min) and recording melting curves. Results were expressed as the normalized ratio of the mRNA level of the target gene to the internal control actin gene (*ACT*). Changes were estimated using the Relative Expression Software Tool (REST 2009) [30].

## Inoculation of pepper fruits

*PC* culture plates flooded in sterile distilled water were refrigerated at 4˚C for 60 min and then left at room temperature for 45 min to motivate the release of zoospores. The zoospore suspension was prepared ($10^6$ /mL) and used for fruit inoculation [31]. After 70 days of soil treatment, 10 μL of zoospores suspension was injected into fruits of each treated plant. Ten μL sterile saline solution was injected into the fruits of control plants as a negative control. The fruits of each treatment were collected at five days after inoculation and disease index (DI) was calculated based on measuring the area of water-soaked tissue [32].

## Statistical analysis

Statistical analysis was performed using analysis of variance (ANOVA) by Microsoft Excel (Microsoft Corporation, USA) and SPSS version 22.0 (SPSS Inc. Chicago, IL USA) packages. All data shown are the average value of three (*in vitro* experiments) biological replicates ± SE. All of the experiments were repeated two times. The greenhouse trials were carried out as a full factorial experiment in randomized blocks design with five blocks and there were five independent biological replicates for each treatment. The significance of differences among treatments

**Table 1. The primers were designed in this study for q-RT PCR.**

| Target gene | Amplicon size (bp) | Sequence |
|---|---|---|
| *GST* | 168 | F-3' GACTTTGGCTGGGAGTTT5'<br>R-5' CTTGGAAACGAGCTTGGA 3' |
| *WRKY53* | 175 | F-5' TTGTCTCTCCTACAACCC 3'<br>R-5' CGTTCAAGTGGAAACTCC 3' |
| *WRKY40* | 165 | F-5' CCAGCCTACTTTCCAACT 3'<br>R-5' CTCCTCAAGACCGCTATC 3' |
| *SUS* | 172 | F-5' CAAGGACAGGAACAAACC 3'<br>R-5' GATTTGGAAGAACAGGCC 3' |
| *PR1* | 184 | F-5' CCAGGTAATTGGAGAGGAC 3'<br>R-5' CTCCAGTTACTGCACCAT 3' |
| *PR10* | 142 | F-5' GCAGATGGAGGATGTGTT 3'<br>R-5' CATACCTCCTCGCCAA 3' |
| *ACCO* | 160 | F-5' AGGAGCCTAGGTTTGAAG 3'<br>R-5' CTCCACACCATTAGCAAC 3' |
| *ERF* | 151 | F-3' ATCTCCACTCCGATTTCC 5'<br>R-5' GGACTGAGGATGTTGTCT 3' |

was evaluated using the multivariate generalized linear model (GLM) with Duncan multiple test post hoc analysis at the level of *P< 0.05*. Heatmap was made by Graph Pad Prism software (version 5, 2005).

## Results

### Selection of anti-Oomycete strains

Among studied strains, four strains showed growth inhibition greater than 50% against *PC* on PDA medium. IT20 and SS14 strains, respectively by 69% and 63% inhibition of mycelial growth (Fig 1) that varied in phosphate solubilizing capability were selected for evaluation in the following experiments. *In vitro* data showed that IT20 and SS14 did not antagonize growth and P-solubilizing activity of each other (S1 Fig). The combination of two species synergistically inhibited (82%) mycelial growth of *PC* compared to individual dual culture with SS14 (40%) and IT20 (50%) (Fig 1).

### Pepper growth promotion

IT20 + SS14 and IT20 significantly increased shoot length (23 and 16%, respectively) compared to control plants after 30 days of treatments (Table 2). Likewise, these two treatments significantly (*p<0.05*) accelerated the flowering transition and promoted the number of flowers compared to control and SS14. There was no significant difference in the number of fruits among bacterial treated plants. After 70 days, IT20 + SS14 significantly increased fruit length by 70 and 22% compared to control and the single application, respectively. Also, the most total fruit weight was associated with IT20 + SS14. IT20 + SS14 significantly enhanced total fruit weight by 76% compared to control plants (Table 2). IT20 followed by SS14 revealed maximum plant weight compared to other treated plants after 80 days. *Streptomyces* did not any increase in root weight compared to control.

### Bio-suppression of pepper plant disease

The disease symptoms of plants were observed seven days after inoculation (dai) of the pathogen. The highest DS was observed at 21 dai (Fig 2A). As shown in Fig 2, there was a significant difference in the levels of DS among bacterial treatments and *PC* at seven days after

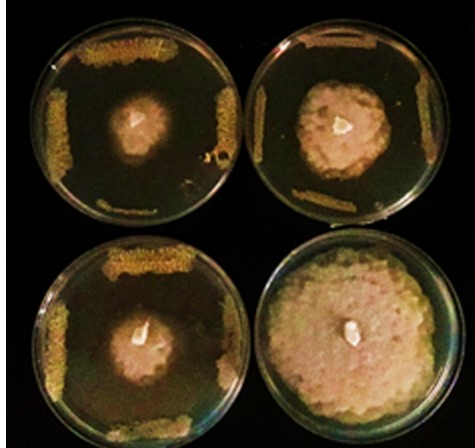 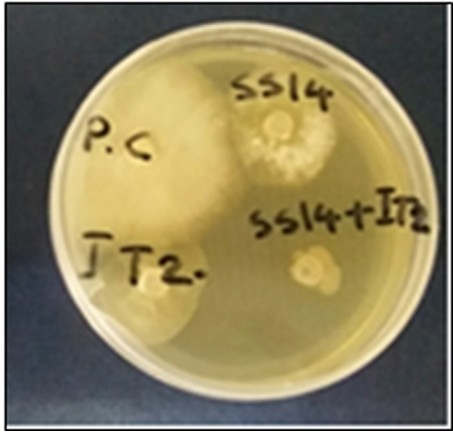

**Fig 1. IT20, SS14, IT20+SS14, and control (PC) in dual culture assay; and inhibition of mycelial growth and germination of *Phytophthora capsici* after six days of incubation.**

**Table 2. Effect of bacterial treatments and beta aminobutyric acid (BABA) on growth and bio-suppression of pepper blight disease caused by *Phytophthora capsici* (*PC*).**

| Treatment | Vegetative growth-promoting | | Reproductive growth-promoting | | Fruit-growth promoting | |
|---|---|---|---|---|---|---|
| | Shoot length 30 dai (cm) | Plant weight 80 dai (g) | No of flowers/plant 30 dai | No of fruits/plant 50 dai | Fruit length 70 dai (cm) | Total fruit weight 80 dai (g) |
| C | 23.2±0.8[d*] | 72.0±7.9[b] | 9.0±0.7[b] | 1.0 [b] | 4.2±0.5[c] | 218.0[b] |
| BABA | 24.2±0.5[cd] | 72.4±5.5[b] | 6.4±2.4[b] | 1.0 [b] | 5.2±0.3[b] | 115.0[d] |
| SS14 | 25.0±0.5[c] | 81.4±15.2[ab] | 10.4±0.5[b] | 1.0 [b] | 5.7±0.6[ab] | 162.0[c] |
| IT20 | 27.6±0.4[b] | 92.2±7.4[a] | 13.4±2.1 [a] | 1.0 [b] | 6.1±0.6[ab] | 243.0[b] |
| IT20+SS14 | 28.6±0.5[a] | 65.8±11.7[b] | 13.2±1.8 [a] | 1.0 [b] | 7.2±0.1[a] | 384.0[a] |
| PC | 19.0±2.1[e] | 40.4±10.7[c] | 8.0±1.4[c] | 1.0 [b] | 3.6±0.6[c] | 153.0[c] |
| BABA+PC | 20.0±2.7[de] | 52.8±8.5[c] | 8.0±1.2[c] | 2.0 [ab] | 6.8±0.6[ab] | 150.0[c] |
| SS14+PC | 22.6±4.1[cde] | 102.2±25.0[a] | 10.0±0.0[b] | 2.0 [ab] | 6.0±0. 9[ab] | 147.0[c] |
| IT20+PC | 25.6±0.8[c] | 52.6±13.8[bc] | 12.4±1.5[a] | 2.0 [ab] | 7.4±0.5[a] | 370.0[a] |
| IT20+SS14 +PC | 27.0±0.6[ab] | 82.8±11.9[ab] | 13.8±1.5[a] | 3.0 [a] | 7.0±0.8[ab] | 223.0[b] |

Values are the means (averaged over five replicates) ± SE

* Same letters represent non-significant difference according to Duncan's Multiple Range Test ($P < 0.05$)

The results of greenhouse experiments on the vegetative and reproductive parameters were recorded from 30–80 days after bacterium treatment

inoculation ($p< 0.05$). The highest level of DS was observed in positive control inoculated with *PC*. A significant difference was found among IT20+SS14, IT20, and SS14 treatments at 21 dai (Fig 2B). Soil treatment with *Streptomyces* species significantly ($p< 0.05$) decreased disease symptoms of blight from 40 to 60% compared to *PC*. Minimum shoot length and plant weight, number of fruits, and fruit length were associated with *PC*. IT20 + SS14 significantly promoted ($p< 0.05$) shoot length by 36% compared to *PC* (Table 2). In inoculated plants, there was a significant difference in the number of flowers between IT20 + SS14 and SS14 after 30 days of bacterial treatment. IT20 promoted fruit length and fruit weight by 100 and 140%, respectively compared to *PC* (S2 Fig). The tissue responses of treated plants modulating pepper systemic resistance were investigated in terms of $H_2O_2$ accumulation and gene expression analysis.

## Hydrogen peroxide accumulation

All BABA, IT20+SS14, IT20, SS14 treated plants, and *PC* increased $H_2O_2$ production at seven dai. Maximum level of $H_2O_2$ accumulation was associated with *PC* in both leaves and fruit tissues. In the leaves of IT20+SS14-treated plants, $H_2O_2$ accumulation was lower compared to other inoculated treatments. Plant inoculation increased the amount of $H_2O_2$ in the fruit tissues of IT20-treated plants (Fig 3).

## Expression pattern of the selected genes in leaf and fruit tissues

There was no significant difference in transcripts of glutathione S-transferase (*GST*) among plants exposed to combination and single treatments after challenging with the pathogen. In inoculated plants, maximum and minimum levels of the relative ratio of *GST* gene were observed in IT20+SS14 and BABA treatments, respectively (Fig 4A). In the leave tissues of BABA- and IT20- treated plants, transcripts of sucrose synthase (*SUS*) were depleted by 107- and 89-fold compared to control, respectively. At seven dai, transcripts of *SUS* significantly were suppressed in the plants inoculated with *PC* and BABA (Fig 4B). SS14 and IT20 significantly stimulated (up-regulated) the expression of pathogenesis-related protein1 (*PR1*) gene. IT20 followed by IT20+SS14 significantly up-regulated *PR1* gene in leaf tissues of inoculated

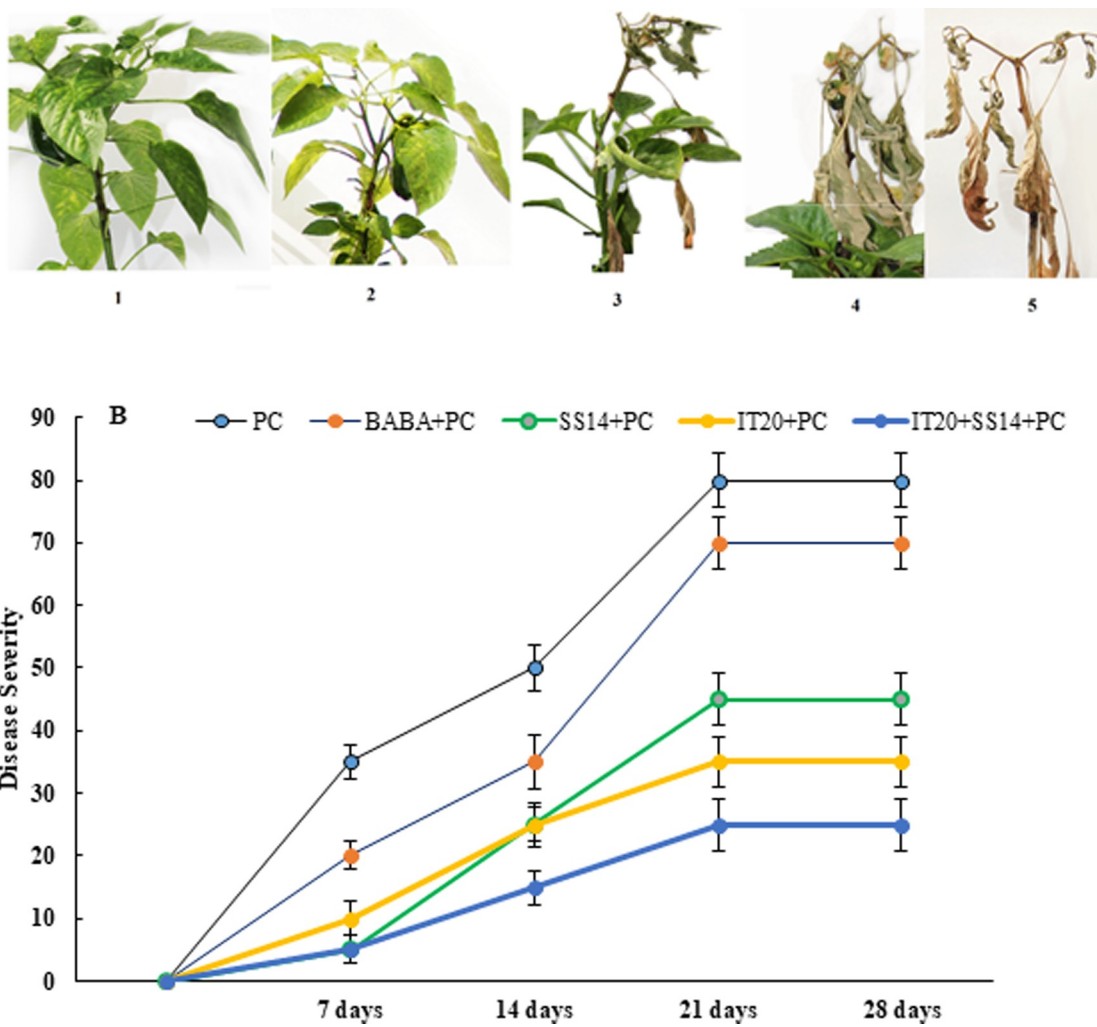

**Fig 2.** Disease scales (**A**) and disease progress (**B**) related to chemical (BABA), biological (*S. vinaceusdrappus* SS14 or *S. rochei* IT20) and combination of biological treatments (IT20+SS14) from 7 to 28 days after inoculation of *Phytophthora capsici* (*PC*).

plants (Fig 4C). In inoculated plants, there was no significant difference in transcripts of *PR10* gene among single and combination treatments (Fig 4D). The results revealed that IT20+SS14 significantly up-regulate transcription factor (TF) *WRKY40* expression by 33- and 22-fold compared to control and *PC*, respectively (Fig 4E). There was a 22- fold up-regulation in the transcription of *WRKY53* in BABA-treated plants after pathogen inoculation (Fig 4F). Transcripts of ET- response factor (*ERF*) significantly increased (50-fold) in *PC* at seven dai (Fig 4G). IT20+SS14 significantly induced transcripts *ERF* gene by 37-fold compared to control, BABA and single treatments. In inoculated plants, the results revealed that IT20+SS14 significantly up-regulated (six-fold) the expression of *ERF* compared to IT20 and SS14 (Fig 4G). IT20+SS14 significantly induced a 50-fold up-regulation in *ACCO* (1-aminocyclopropane-1-carboxylic acid (ACC) oxidase) transcripts compared to control. Pathogen inoculation significantly suppressed transcripts of *ACCO* (Fig 4H). In fruit tissues, transcripts of *GST* significantly were suppressed in all treatments and the maximum relative ratio was associated with IT20+SS14 treatment. The transcripts of *GST* significantly increased in inoculated treatment of IT20. The most suppression of *GST* expression was observed in *PC* (Fig 5A). The transcript

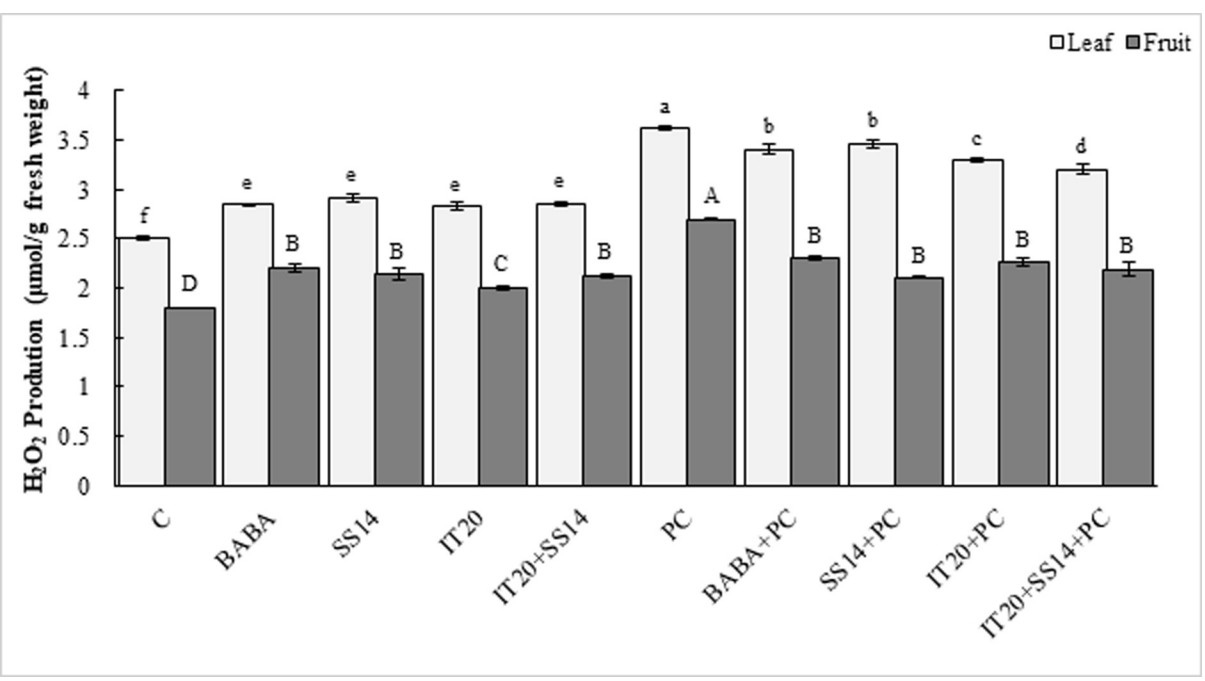

**Fig 3. H₂O₂ production in leaves and fruits of chemical (BABA), biological (*S. vinaceusdrappus* SS14 or *S. rochei* IT20) and combination (IT20+SS14) treated plants after seven days of pathogen inoculation respectively.** Same letters represent non-significant difference according to Duncan's Multiple Range Test ($P < 0.05$).

levels of *SUS* in *PC* and IT20+SS14 decreased by seven- and six-fold compared to control, respectively. The most expression ratio of *SUS* (40-fold) was associated with IT20-treated plants (Fig 5B). In inoculated plants, there was no significant induction of *PR1* transcription among treatments (Fig 5C). Transcripts of *PR10* significantly decreased in *PC* (Fig 5D). The highest expression level of *WRKY40* gene (15- fold) was detected in IT20 followed by IT20+-SS14-treated plants (nine-fold). There was a significant up-regulation of *WRKY40* (80-fold) in the inoculated treatments of IT20 (Fig 5E). The significant up-regulation of *WRKY53* gene was observed in IT20+SS14 (100-fold) and *PC* (69-fold) (Fig 5F). IT20 stimulated the expression of *ERF* (12-fold). In the other pathogen inoculated treatments, the transcripts of *ERF* significantly were suppressed (Fig 5G). There was a significant down-regulation of *ACCO* gene in *PC* (300-fold) followed by IT20+SS14 (96-fold). On the contrary, IT20 up-regulated *ACCO* gene expression by 77-fold compared to control (Fig 5H).

### *Phytophthora* fruit blight assay in treated plants

After five days of zoospore injection into pepper fruits of treated plants, the greatest value of DI was observed in *PC* (Fig 6). The level of DI was lower in the fruits of IT20-treated plants followed by SS14 inoculated plants compared to other treatments. In contrast, the percent of DI in IT20+SS14 was the same as BABA-treated plants.

## Discussion

Mineral nutrition interferes with fungal disease progress by the direct effect on the pathogen, plant growth and development, and resistance mechanisms [33]. Recently, Sang et al. [34] reported a P-solubilizing PGPR strain of *Chryseobacterium* sp. that suppressed *Phytophthora* blight disease and increased yield of pepper plants in the field condition. The efficient root

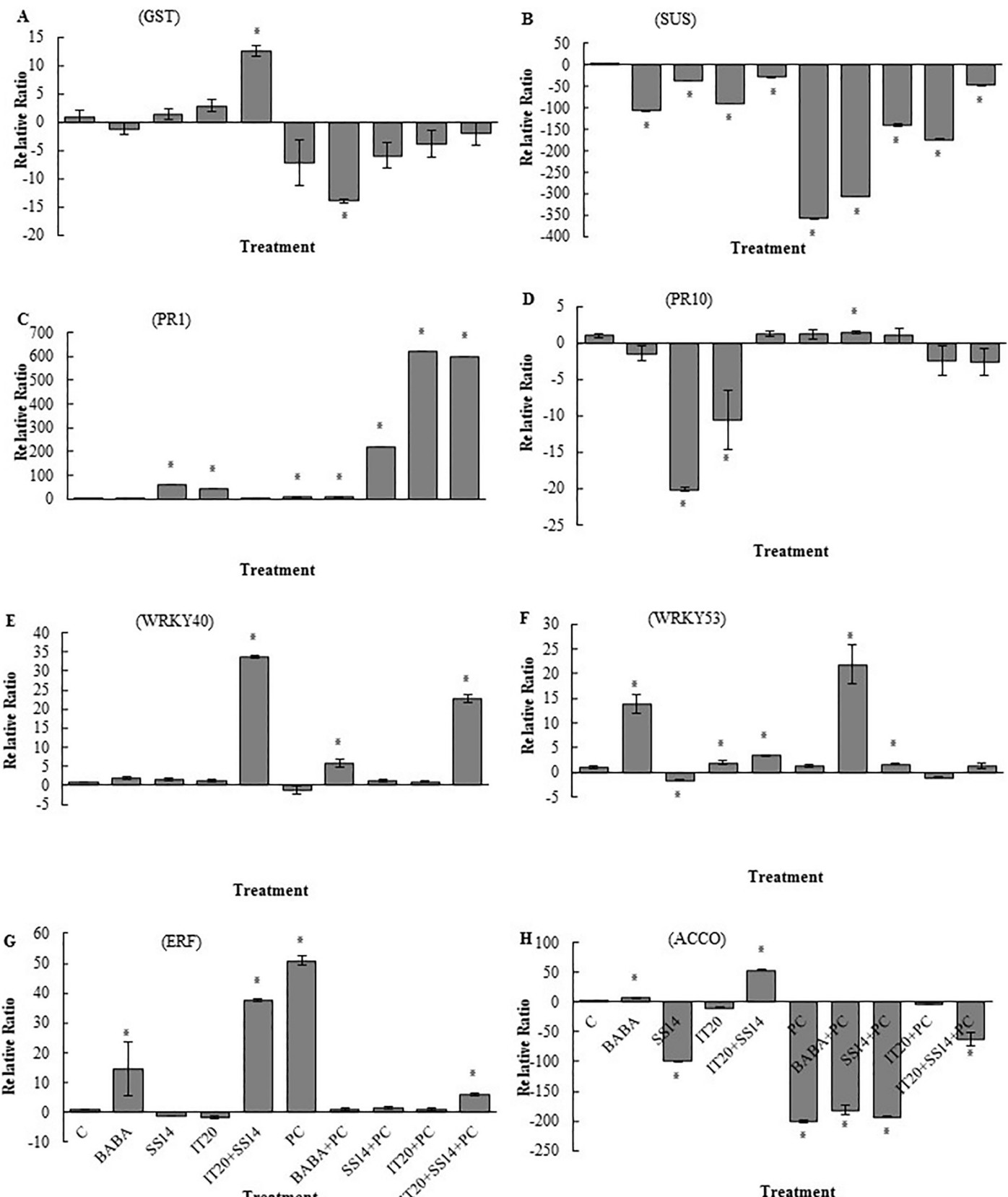

**Fig 4.** The relative level of gene expression (fold) determined by qRT-PCR of eight target genes including *Glutathione-S-transferase* (**A**), *Sucrose synthase* (**B**), *PR1* (**C**), *PR10* (**D**), *WRKY40* (**E**), *WRKY53* (**F**), *ERF* (**G**), and *ACC oxidase* (**H**) versus reference control (*Actin* gene) in leave tissues of pepper treated with chemical (BABA), biological (*S. vinaceusdrappus* SS14 or *S. rochei* IT20) and combination (IT20+SS14) after seven days inoculation with PC/ PDA plugs. Untreated and non-inoculated plants considered as control (C). Standard error represents for three biological replicates. Positive values of fold change indicate up-regulation while negative values have been known for the down-regulated genes. The values marked with an asterisk are significantly different from control at *P< 0.05*.

colonization and growth stimulation mediated by this strain were associated with increasing pepper biomass. In this sense, nutrient availability such as phosphate in the rhizosphere of the treated plants could increase resistance upon pathogen inoculation. Some of PGPRs via some mechanisms could contribute to the sensitization of the plant immune system to overcome attacking pathogens [35]. The phosphate solubilizing *Streptomyces* species might be as ISR triggered PGPRs suggested by Walters. Siderophore-producing *Streptomyces* strains SS14 and IT20 were positive for cellulase activity [7]. As seen from the previous study [36], growth inhibition failure of *Oomycete* pathogen was correlated to the inability of *Streptomyces* strains in producing cellulase. P-solubilizing strain IT20 showed maximum growth inhibition against *PC*. Interestingly, co-inoculation of IT20 and SS14 imposed more inhibition (> 80%) than those with single inoculations (Fig 1). IT20+SS14 significantly suppressed disease severity by 75% and 50% compared to *PC* and SS14, respectively at 28 dai. Indeed, IT20 in combination with SS14 provided similar or greater disease suppression and reproductive growth promotion than IT20 *(*Fig 2*)*. In the same way, Santiago et al. [37] showed that the combination of two *Streptomyces* strains R170 with R182 synergistically increased potato plants weight compared to control, demonstrating the compatibility of strains based on the increased production of plant growth sponsors such as indole-3-acetic acid (IAA) and siderophore along with colonization on roots. Prior findings demonstrated that *Streptomyces* sp. strain AcH505 by producing specific metabolite such as auxofuran could be as a mycorrhiza helper in plant interaction and enhance colonization of plant tissues [38]. To our knowledge, this is the first time that an inorganic P-solubilizing *S. rochei* IT20 in the presence of *S. vinaceusdrappus* SS14 is regarded as a helper and might effectively enhance systemic resistance of pepper plants against *PC*. $H_2O_2$ produced in inoculated plants during oxidative stress has a dual function. It may lead to programmed cell death but also act as a fast signal for stimulation of antioxidative defenses [39]. In IT20+SS14-treated plants, $H_2O_2$ accumulation and transcript level of *GST* gene were lower compared to other inoculated plants, suggesting a slight balance between $H_2O_2$ generation and $H_2O_2$ scavenging (Fig 4A). Also, a lower level of *PR1* expression was observed. These induced responses might increase root colonization rate by suppressing oxidative burst. The plant cells must reduce ROS level to avoid $H_2O_2$ accumulation as an inducer of oxidative damage. In the same way, Lehr et al. [40] designated that mycorrhiza helper bacterium *Streptomyces* sp. AcH 505 can increase root colonization ratio by positive effects on root receptivity. According to their study, such a response was generated by down-regulation of peroxidase activity and pathogenesis-related peroxidase gene expression to suppress the plant defense responses. SUS is a glycosyltransferase involved in sucrose metabolism [41]. In addition to typical roles as carbon source, sucrose acts as a candidate signal molecule to various regulatory mechanisms related to growth, development, response to biotic and abiotic stress factors [42, 43]. In the current study, *SUS* transcript levels were reduced in leaf tissues responding to *PC*, but IT20+SS14 alleviated this deficiency, which suggested there is a clear advantage for pepper plants during pathogen challenge (Fig 4B). In other words, *Streptomyces* combination treatment might sensitize the perception of stress signals by the plant via sucrose synthases. Increased expression of this gene in leaf and fruits of IT20-treated plants represents the role of *SUS* in ISR (Fig 5B). It seems that *SUS* is an appropriate candidate for priming against later attack of *PC* in fruit tissues. In inoculated plants, the up-regulation of *PR1* was routinely used as a marker gene for

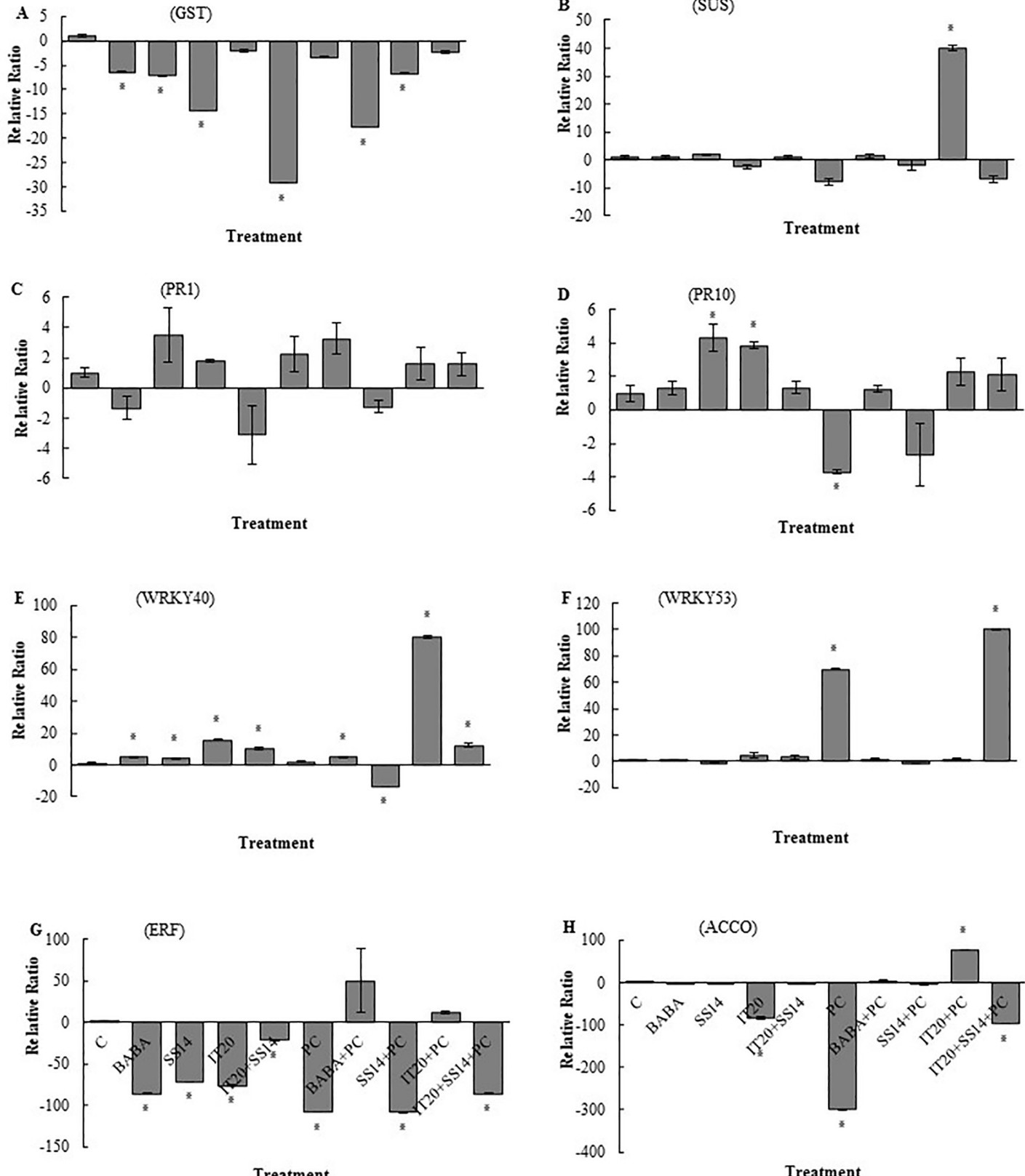

**Fig 5.** The relative level of gene expression (fold) determined by qRT-PCR of eight target genes including *Glutathione-S-transferase* (**A**), *Sucrose synthase* (**B**), *PR1* (**C**), *PR10* (**D**), *WRKY40* (**E**), *WRKY 53* (**F**), *ERF* (**G**), and *ACC oxidase* (**H**) versus reference control (*Actin* gene) in fruit tissues of pepper treated with chemical (BABA), biological (*S. vinaceusdrappus* SS14 or *S. rochei* IT20) and combination (IT20+SS14) after seven days inoculation with PC/ PDA plugs. Untreated and non-inoculated plants considered as control (C). Standard error represents for three biological replicates. Positive values of fold change indicate up-regulation while negative values have been known for the down-regulated genes. The values marked with an asterisk are significantly different from control at *P< 0.05*.

systemic acquired resistance (SAR) and SA-dependent pathway in pepper plants against *P. capsici* [16,44]. Increased abundance of *PR1* transcripts by IT20+SS14 compared to IT20 and SS14 just in inoculated plants indicated that these treatments probably stimulated different levels of defense priming in pepper plants. Oelke et al. [45] designated different levels of resistance existing to foliar blight, stem blight, and fruit rot of pepper. Defense responses of pepper leaves may be induced more rapidly and strongly than in fruits due to the induction of *PR1* gene expression in leaves inoculated with *PC*. Taken together, IT20+SS14 treatment was probably effective to establish and develop priming in leaves of pepper plants. Priming is a process that prepares a plant for stronger and faster defense activation before stress conditions [46]. The relative expression of other candidate genes encoding TFs (*WRKY40*, *WRKY53*, and

**A**

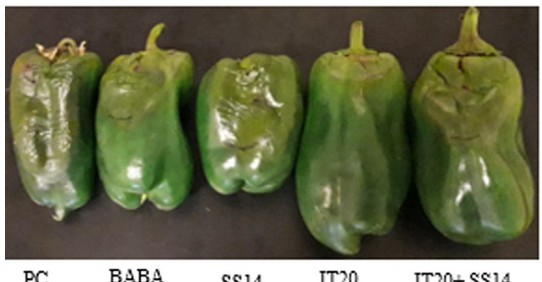

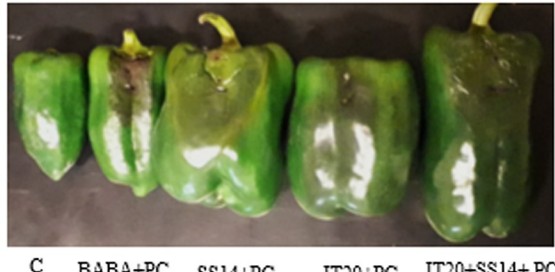

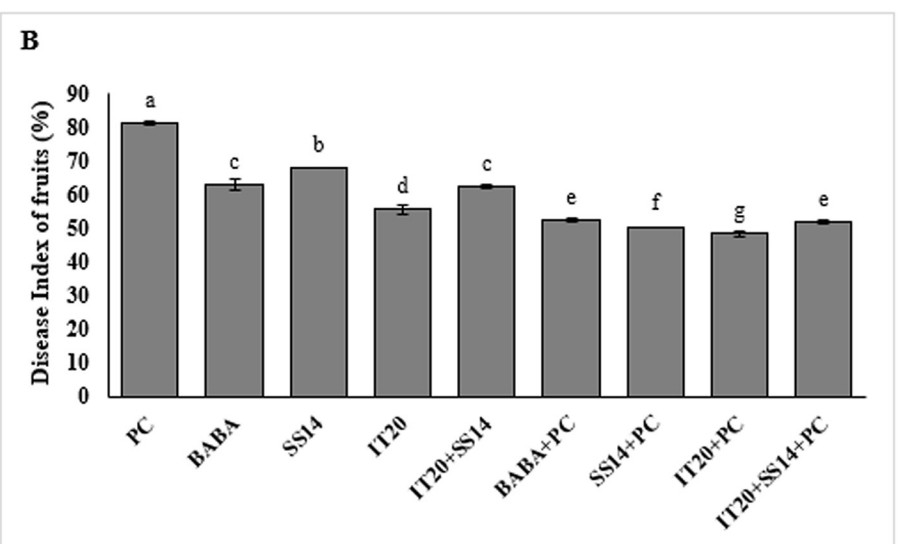

**Fig 6.** Disease severity (**A**) and disease index (**B**) of fruits inoculated with the zoospore suspension of *Phytophthora capsici* (5*10⁶/ml) on pre-inoculated (left) and non-inoculated (right) after five days of inoculation. The same letters represent non-significant difference according to Duncan's Multiple Range Test (*P < 0.05*). PC: inoculated control and C: non-inoculated control.

*ERF*), and *ACCO* was evaluated in the following experiments. In pepper, *WRKY40* was proved to be involved in the modulation of resistance *against Ralstonia solanacearum* [47, 48]. In *leaf tissues* of IT20+SS14 treated plants, up-regulation of *WRKY40* gene expression along with the increase in fruit size expression among co-inoculation and individual species (Fig 4E). Similarly, plant inoculation with *PC in pre-treatments of IT20 significantly increased the transcripts of WRKY40 in the fruit tissues*. These data supported the role of *WRKY40* as a positive regulator of defense priming in pepper against *PC*. Conversely, *WRKY53 in* leaf and fruit tissues **acts as a negative** regulator of defense response against *PC* in **constitutive expression with** *WRKY53*. The findings revealed that *WRKY40 not only contribute to inducing resistance but also may be involved in the* fruit ripening *process*. Our results are following the study of Chen et al. [49] indicated differences in *WRKY* gene expression among mutants and provided insights into their probable roles in ripening the fruit and strong induction of *WRKY* genes in stress condition. Inhibiting expression reduction of *ERF* upon pathogen attack of IT20+-SS14-treated plants along with reduced $H_2O_2$ accumulation suggested that *ERF* has a positive regulatory role in defense responses against *PC* in the leave tissues. The results of the expression profile showed this gene could not only respond to the infection of *P. capsici* but also be induced by BABA. In a similar pattern, the results of real-time PCR analysis was done by Jin et al. [50] showed that ERF gene family of pepper had the high expression level and could respond to the infection of *PC* and the signaling molecules (SA, Methyl Jasmonate, Ethephon, and hydrogen peroxide). In a previous study, co-inoculation of *Pseudomonas* sp. as a PGPR with mycorrhizal fungus *Rhizophagus irregularis* induced activation of defense-related genes of potato plants against *R. solani* and greater ISR than single application via priming of ET resistance network [51]. In the current study, changes in the expression of *ACCO* and *ERF* genes in two tissues of the plant have the same direction. This may be related to the role of *ACCO* in ET-related signal transduction after post-translational processing. A previous study revealed *ACCO* stimulates phosphorylation of apple fruit proteins in the ripening-dependent changes [52]. Altogether, ACC (the precursor of ET) and ET could act as the signaling molecules under stressful conditions and the high level of *ERF* gene expression could be efficient response led to decrease high levels of ACC and ET in leaf tissues. In the same way, Guan et al. [53] revealed that *Arabidopsis* mutants in *ACC Synthase* (defect in ET production) showed greater susceptibility to *P. syringae* infection. The observations proposed that *ERF* gene in leaf tissue was more strongly up-regulated in IT20+SS14 than plants only inoculated with IT20. *ACCO* gene expression induced by single treatment of IT20 strongly up-regulated upon challenge with the pathogen in fruit tissues (Figs 4 and 5). Our data support this hypothesis that different IT20 (singly and in combination) treatments induced different regulation of ET-dependent pathway, displaying *ERF* and *ACCO* are the suitable responses for regulating of ET-dependent pathway respectively in leaves and fruits of pepper against *PC*. IT20+SS14 in treated plants accelerated flowering time compared to control plants; the pathogen inoculated plants promoted much more stimulation than non-inoculated. Some studies have suggested that biotic stress factors could play roles in controlling the flowering [54]. This view could be associated with higher abundance of *SUS* and *ERF* transcript levels that accelerated flowering and juvenile-to-adult phase transition [55, 56]. Age-related resistance, an event whereby mature plants become more resistant to some pathogens, has been associated with the shifting to flowering [57]. The pepper cultivar used in this experiments was susceptible to fruit blight disease caused by *PC*. The fruits of IT20-treated plants provided a greater level of resistance than SS14 (3%) and IT20+SS14-treated plants (6%) (Fig 6). Interestingly, the infected fruits collected from inoculated plants showed a lower level of DI than non-inoculated treatments. Plant inoculation with *PC* significantly boosted the accumulation of $H_2O_2$ in the fruits of IT20-treated plants. Likewise in the results of Esmaeel et al. [58], the accumulation of $H_2O_2$ in *Burkholderia*

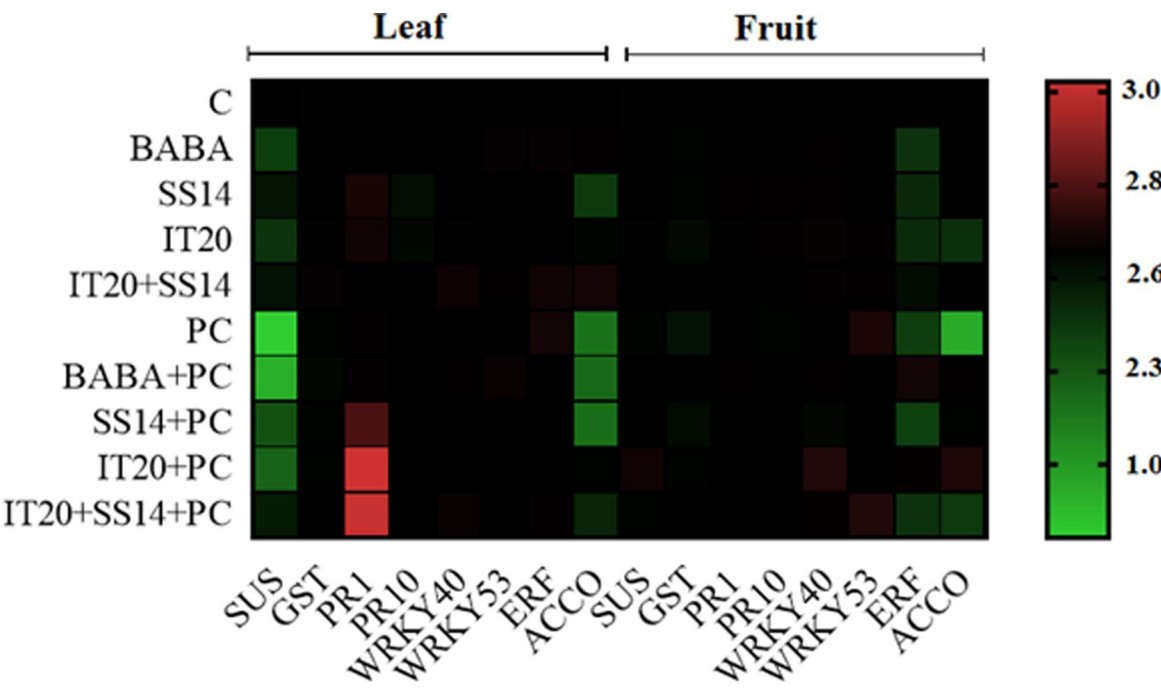

**Fig 7. The tissue-specific expression of target genes of *C. annuum* visualized using *log-2* fold change values in a heat map.** The plants were treated with the chemical (BABA), biological (*S. vinaceusdrappus* SS14 or *S. rochei* IT20) and combination (IT20+SS14) after inoculation with PC. Untreated and non-inoculated plants considered as control (C).

treated-grapevine increased after the inoculation of *B. cinerea*. Upon pathogen attack, these biocontrol strains induced ROS generation which in turn led to priming the defense of disease. Although there was a significant difference in stem and leaf blight disease suppression among co-inoculated and single treatments, this cannot be considered as *Phytophthora* fruit blight resistance induced by IT20+SS14; because of the development of the disease symptoms in fruits of treated plants was higher than a single treatment. The results are accordance with Jambhulkar et al. [59] indicated the performance of a combination of biocontrol agents in comparison with single application depend on the targeted disease, as observed in rice plants for the combination of *Trichoderma* and *Pseudomonas* strains, which was more effective than single against blast disease but not against bacterial leaf blight. This is the first report of changes in expression of some defense-related genes in different tissues of the host plant can play role in tissue-dependent synergistic induced systemic resistance by a combination of two *Streptomyces* species against *PC* (Fig 7). Our findings provide insights into systemic resistance induced by *Streptomyces* depend on not only co-regulation of expression genes involved in SA- and ET-dependent pathways, but also tissue-specific defense responses of the host plant affected during the interactions. The use of synergist antagonists to modulate plant defense responses against the pathogen undoubtedly needs further studies and a deep understanding of regulatory mechanisms of responsive target tissues.

## Conclusion

The combination of two *S. vinaceusdrappus* SS14 with *S. rochei* IT20 significantly inhibited mycelial germination of *P. capsici*. This new combination significantly depleted blight severity caused by the pathogen on pepper plants through ISR and priming-mediated systemic resistance. IT20 promoted fruit growth and induced systemic resistance with a primed state

probable via the induction of *ACCO* and *SUS* gene against the later infection of the pathogen in the fruits. Identification of the expression profiles among different treatments and plant tissues should help to enhance the efficacy of priming in plants. Such findings are crucial, as they offer insights to study the mechanisms and the interactions between the plant tissue genetic background and priming. Furthermore, the generation of knowledge on the combination of candidate inducers and gene expression results could move toward breeding programs to reach more durable and sustainable plant protection.

## Supporting information

**S1 Fig.** IT20 (right) and SS14 (left) on ISP2 (A) and inorganic phosphate solubilizing medium after six days of incubation (B). IT20 shows inorganic phosphate solubilizing activity. These species have not inhibition interaction shows that the combination of IT20 and SS14 is compatible.
(DOCX)

**S2 Fig. Shoot and roots of pots inoculated with *Phytophthora capsici* after 80 days of bacterial treatments.**
(DOCX)

## Author Contributions

**Conceptualization:** Naser Safaie, Akram Sadeghi.

**Data curation:** Naser Safaie.

**Formal analysis:** Sakineh Abbasi, Naser Safaie.

**Funding acquisition:** Akram Sadeghi.

**Investigation:** Sakineh Abbasi.

**Methodology:** Naser Safaie.

**Project administration:** Akram Sadeghi.

**Supervision:** Naser Safaie, Akram Sadeghi.

**Writing – original draft:** Sakineh Abbasi.

**Writing – review & editing:** Naser Safaie, Akram Sadeghi, Masoud Shamsbakhsh.

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
