## [Decision Letter · Decision Letter 0]

6 Dec 2019

PONE-D-19-30373

Tissue-specific synergistic bio-priming of pepper by two Streptomyces species against Phytophthora capsici

PLOS ONE

Dear Dr Naser Safaie,

Thank you for submitting your manuscript to PLOS ONE. After careful consideration by unfortunately a single reviewer (10 were approached), we feel that it has merit but does not fully meet PLOS ONE’s publication criteria as it currently stands. Therefore, we invite you to submit a revised version of the manuscript that addresses the variouspoints raised by the reviewer, that should help you to improve your manuscript. 

Be aware that I will try to find other reviwers besides this unique  reviewer to juge your revised manuscript.

We would appreciate receiving your revised manuscript by mid january 2020. To enhance the reproducibility of your results, we recommend that if applicable you deposit your laboratory protocols in protocols.io, where a protocol can be assigned its own identifier (DOI) such that it can be cited independently in the future. For instructions see: http://journals.plos.org/plosone/s/submission-guidelines#loc-laboratory-protocols

We look forward to receiving your revised manuscript.

Kind regards,

Marie-Joelle Virolle, PhD

Academic Editor

PLOS ONE

Journal Requirements:

2. Please include your tables as part of your main manuscript and remove the individual files. Please note that supplementary tables (should remain/ be uploaded) as separate "supporting information" files

Additional Editor Comments (if provided):

Reviewers' comments:

Reviewer's Responses to Questions

**Comments to the Author**

1. Is the manuscript technically sound, and do the data support the conclusions?

Reviewer #1: Partly

2. Has the statistical analysis been performed appropriately and rigorously? 

Reviewer #1: Yes

3. Have the authors made all data underlying the findings in their manuscript fully available?

Reviewer #1: Yes

4. Is the manuscript presented in an intelligible fashion and written in standard English?

Reviewer #1: No

5. Review Comments to the Author

Reviewer #1: Comments on the manuscript: “Tissue-specific synergistic bio-priming of pepper by two Streptomyces species against Phytophthora capsica”

This manuscript reports studies on protection of pepper plant from the fungal plant pathogen, Phytophthora capsici. The investigators have examined in-vitro inhibition of the pathogen by the selected strains of Streptomyces spp. Subsequently they have the experimental pepper plant with the Streptomyces spp. and challenged them with the pathogen. Parameters of the plant, like, shoot length, fruit length and weight etc. were measured as an indication of the health condition of the plant. Further, effect of Streptomyces strains on treated plants in respect of accumulation of H2O2 and transcription level of defense related genes were examined.

While the subject of study is interesting, the paper is not suitable for publication in its present form. The authors require to address the following queries before the manuscript can be considered further.

1. The manuscript needs major improvement in English language. It should re-written for clarity and accuracy of language.

2. Name of the organisms must be in italicized throughout the text.

3.The method section does not provide clear description of experimental procedure. This needs to be appropriately revised.

4. Please clarify how cfu/ml suspension of Streptomyces strains were achieved by using cells and spores?

5. Whether the Streptomyces strains were able to colonize the root, leaf or fruit of the plant following treatment with the culture?

6. What was the level of expression of plant defense genes during the 7-days period following treatment of the plant with the Streptomyces spp. but before inoculation of the P. capsici into the plant?

7. Table 1 seems unnecessary and should be removed. Necessary information may be included in the materials and methods.

8. There is no description of Fig. 1 in the text??

9. Superscripts in Table 3 should be explained in note under the table.

10. What was the rational for measuring shoot length 30 days after inoculation and plant weight on 80 dai? Whether these measurements were done only once during the entire period of experiment?

11.Results pertaining to Phytophthora fruit blight assay of treatments, contained in Fig. 6 should be moved at the end of the result section after description of Fig 5.

12. Since several Streptomyces spp. are reported to induce systemic acquired resistance in plants and protect them against phytopathogenic fungi, what is novel in this study?

6. PLOS authors have the option to publish the peer review history of their article (what does this mean?). If published, this will include your full peer review and any attached files.

Reviewer #1: Yes: Ashok K. Dubey

---

## [Author Response · Author response to Decision Letter 0]

14 Jan 2020

Authors are grateful for reviewing the manuscript and good comments. We tried to improve and correct the manuscript according to the reviewer's comments. The answers (R) to the questions (Q) are as follow:

Q1. The manuscript needs major improvement in English language. It should re-written for clarity and accuracy of language.

R. Major improvements and changes were made in the manuscript, grammatical errors were corrected and conclusion was rewritten.

Q2. Name of the organisms must be in italicized throughout the text.

R. The name of the organisms italicized throughout the text.

Q3.The method section does not provide clear description of experimental procedure. This needs to be appropriately revised. 

Q4. Please clarify how cfu/ml suspension of Streptomyces strains were achieved by using spores?

R. Method section was revised and clearly explained. We suppose all the questions are answered accurately. We cultivated one plug of seven days old bacterial strains in ISP2 media supplemented with 10 mM NaCl, then the spores were counted with hemocytometer. 

Q5. Whether the Streptomyces strains were able to colonize the root, leaf or fruit of the plant following treatment with the culture? 

R. Indeed, we only noted the effect of soil treatment by these two species and reported that Streptomyces were able to colonize the root plant following re-isolation and culturing. We did not examine leaf and fruit colonization.

Q6. What was the level of expression of plant defense genes during the 7-days period following treatment of the plant with the Streptomyces spp. but before inoculation of the P. capsici into the plant?

R. Just, soil drench with chemical inducer BABA (10mM) was performed 48 h before P.capsici inoculation. After seven days of bacterial treatment (to establish), plants were inoculated with the pathogen. The first disease symptoms were observed on pepper plants seven days after pathogen inoculation. So, after seven days of P.capsici inoculation leaves and fruits (by length 1 cm) were harvested for study. We have two groups of treatments inoculated and non-inoculated with pathogen. 

Q7. Table 1 seems unnecessary and should be removed. Necessary information may be included in the materials and methods.

R. Table 1 was removed and included in the content of material and methods anywhere needed.

Q8. There is no description of Fig. 1 in the text??

Q9. Superscripts in Table 3 should be explained in note under the table.

R. subscription into note and description of Fig 1 were done.

Q10. What was the rational for measuring shoot length 30 days after inoculation and plant weight on 80 dai? Whether these measurements were done only once during the entire period of experiment?

All of them were conducted during 30-80 days after treatment in two time repeats of experiments. We observed difference in plant height promotion after 30 days of treatment. At the end of study we observed some treatments have less reproductive promotion but more vegetative growth promotion, so we measured plant weight.

Q11. Results pertaining to Phytophthora fruit blight assay of treatments, contained in Fig. 6 should be moved at the end of the result section after description of Fig 5.

R. The results of fruit rot assay was moved after Fig 5.

Q12. Since several Streptomyces spp. are reported to induce systemic acquired resistance in plants and protect them against phyto-pathogenic fungi, what is novel in this study?

The results showed that phosphate solubilizing Streptomyces strain has the potential to induce systemic resistance and reduce the effects of leaf blight disease in compatible combinations in the greenhouse but expression analysis in two different tissues was distinct. Such findings are important, as they open doors to study the mechanisms and the interactions between the genetic background and priming in different tissues of the host plant.

---

## [Decision Letter · Decision Letter 1]

7 Feb 2020

PONE-D-19-30373R1

Tissue-specific synergistic bio-priming of pepper by two Streptomyces species against Phytophthora capsici

PLOS ONE

Dear Dr Naser Safaie,

Thank you for submitting your manuscript to PLOS ONE. After careful consideration by unfortunately a single reviewer, we feel that it has merit but does not fully meet PLOS ONE’s publication criteria as it currently stands. Therefore, we invite you to submit a revised version of the manuscript that addresses the various points raised during the review process. Be aware that efforts would be made by the editor to find more than one reviewer for your revised version.

We would appreciate receiving your revised manuscript by end of february. To enhance the reproducibility of your results, we recommend that if applicable you deposit your laboratory protocols in protocols.io, where a protocol can be assigned its own identifier (DOI) such that it can be cited independently in the future. For instructions see: http://journals.plos.org/plosone/s/submission-guidelines#loc-laboratory-protocols

We look forward to receiving your revised manuscript.

Kind regards,

Marie-Joelle Virolle, PhD

Academic Editor

PLOS ONE

Reviewers' comments:

Reviewer's Responses to Questions

**Comments to the Author**

1. If the authors have adequately addressed your comments raised in a previous round of review and you feel that this manuscript is now acceptable for publication, you may indicate that here to bypass the “Comments to the Author” section, enter your conflict of interest statement in the “Confidential to Editor” section, and submit your "Accept" recommendation.

Reviewer #1: (No Response)

2. Is the manuscript technically sound, and do the data support the conclusions?

Reviewer #1: Yes

3. Has the statistical analysis been performed appropriately and rigorously? 

Reviewer #1: Yes

4. Have the authors made all data underlying the findings in their manuscript fully available?

Reviewer #1: (No Response)

5. Is the manuscript presented in an intelligible fashion and written in standard English?

Reviewer #1: No

6. Review Comments to the Author

Reviewer #1: Major revision of English language is still required. Authors should highlight the changes made in the revised version for a quick reference and review.

7. PLOS authors have the option to publish the peer review history of their article (what does this mean?). If published, this will include your full peer review and any attached files.

Reviewer #1: No

---

## [Author Response · Author response to Decision Letter 1]

21 Feb 2020

Dear Marie-Joelle Virolle,

I am grateful for reviewing the manuscript and good comments. We tried to improve and correct the manuscript according to the reviewer's comments and the revised manuscript is submitted.

C2. The manuscript needs major improvement in English language. It should re-written for clarity and accuracy of language.

R. Major improvements and changes were made in the manuscript, grammatical errors were corrected and conclusion was rewritten.

Best regards,

Naser Safaie

---

## [Decision Letter · Decision Letter 2]

3 Mar 2020

Tissue-specific synergistic bio-priming of pepper by two Streptomyces species against Phytophthora capsici

PONE-D-19-30373R2

Dear Dr. Naser Safaie,

We are pleased to inform you that your manuscript has been judged scientifically suitable for publication and will be formally accepted for publication once it complies with all outstanding technical requirements.

With kind regards,

Marie-Joelle Virolle, PhD

Academic Editor

PLOS ONE

Additional Editor Comments (optional):

Reviewers' comments:

Reviewer's Responses to Questions

**Comments to the Author**

1. If the authors have adequately addressed your comments raised in a previous round of review and you feel that this manuscript is now acceptable for publication, you may indicate that here to bypass the “Comments to the Author” section, enter your conflict of interest statement in the “Confidential to Editor” section, and submit your "Accept" recommendation.

Reviewer #1: All comments have been addressed

2. Is the manuscript technically sound, and do the data support the conclusions?

Reviewer #1: Yes

3. Has the statistical analysis been performed appropriately and rigorously? 

Reviewer #1: Yes

4. Have the authors made all data underlying the findings in their manuscript fully available?

Reviewer #1: Yes

5. Is the manuscript presented in an intelligible fashion and written in standard English?

Reviewer #1: Yes

6. Review Comments to the Author

Reviewer #1: Queries and concerns raised during the review process have been addressed adequately by the authors. The revised version of the manuscript can now be accepted for publication.

7. PLOS authors have the option to publish the peer review history of their article (what does this mean?). If published, this will include your full peer review and any attached files.

Reviewer #1: Yes: Ashok K. Dubey,

---

## [Editor Report · Acceptance letter]

5 Mar 2020

PONE-D-19-30373R2 

Tissue-specific synergistic bio-priming of pepper by two *Streptomyces* species against *Phytophthora capsici*

Dear Dr. Safaie:

I am pleased to inform you that your manuscript has been deemed suitable for publication in PLOS ONE. Congratulations! Your manuscript is now with our production department. 

With kind regards,

on behalf of

Dr. Marie-Joelle Virolle 

Academic Editor

PLOS ONE